**Cite this article:** van Dam A. 2019 Diversity and its decomposition into variety, balance and disparity. *R. Soc. open sci.* **6**: 190452.

complexity/ecology/biogeography

Hill numbers, α-diversity, β-diversity, entropy, aggregation, mutual information

**Author for correspondence:**
Alje van Dam
e-mail: a.vandam@uu.nl

# Diversity and its decomposition into variety, balance and disparity

## Alje van Dam

Copernicus Institute of Sustainable Development and the Centre for Complex Systems Studies (CCSS), Utrecht University, Utrecht, The Netherlands

 AvD, 0000-0002-3321-6399

Diversity is a central concept in many fields. Despite its importance, there is no unified methodological framework to measure diversity and its three components of variety, balance and disparity. Current approaches take into account disparity of the types by considering their pairwise similarities. Pairwise similarities between types may not adequately capture total disparity, since they do not take into account in which way pairs are similar. Hence, pairwise similarities do not discriminate between similarities of types in terms of the same feature and similarities in which all pairs share different features. This paper presents an alternative approach which is based on the overlap of features over the whole set of types. This results in a measure of diversity that takes into account the aspects of variety, balance and disparity. Based on this measure, the '*ABC* decomposition' is introduced, which provides separate measures for the variety, balance and disparity, allowing them to enter analysis separately. The method is illustrated by analysing the industrial diversity from 1850 to present while taking into account the overlap in occupations they employ. Finally, the framework is extended to take into account disparity considering multiple features, providing a helpful tool in analysis of high-dimensional data.

## 1. Introduction

Diversity is a central concept in a wide range of scientific fields. In the natural sciences, it is often associated with the functional properties of a system, like the stability of ecosystems [1,2]. In the social sciences, the concept of diversity is key to theories regarding recombinant innovation [3,4], regional development [5], cultural evolution [6] and the science of science [7–9].

But what exactly is diversity and how can it be measured? Recent frameworks emphasize that diversity consists of three dimensions [10–13]. First, the *variety* describes the number of different types,

species or categories present.[1] The variety is bounded by the total number of types in the classification or taxonomy that is used. Second, the *balance* describes how individuals or elements are distributed across these types. When elements are concentrated in few types the balance is low, while a high balance indicates a more even distribution. Last, the *disparity* takes into account to what extent the types considered differ from each other in terms of some given features or characteristics. If the types considered are very similar, they have low disparity. An increase along any of these three dimensions corresponds to an increase in overall diversity. A proper measure of diversity should therefore take into account all three dimensions.

Despite the importance of diversity as a concept, there is no unified methodological framework to measure and analyse the three dimensions of diversity. In the past, the disparity was not even considered by most diversity indices. There have been multiple attempts to incorporate disparity into a measure of diversity by including some measure of the pairwise distances or similarities between the types considered. An example is Rao's quadratic entropy [14], introduced into the social sciences in [13], where it is known as the Rao–Stirling diversity. It expresses diversity as the average distance between types, weighed by their relative frequencies.

More recently, it has been shown in [15] that Rao's quadratic entropy follows as a special case from a more general framework that generalizes the so-called Hill numbers [16,17] to include pairwise similarities between types. A similar approach was taken in [18], which generalizes Hill numbers to include phylogenetic or functional similarities. Both approaches compute diversity in terms of 'effective numbers' [17], based on a distribution of types and a given matrix that contains the pairwise similarities between those types. Other approaches to quantify diversity while taking into account disparity have been to compose a measure of diversity by separately measuring variety, balance and disparity, and combining them into a single index of diversity [19]. A more data-driven approach is taken in [8], which applies factor analysis to a range of different diversity indices to infer three variables that correspond to variety, balance and disparity.

What all approaches described above have in common is that disparity is quantified using pairwise similarities. The use of pairwise similarities, however, may lead to both practical and conceptual problems. One practical problem is that there are many different ways in which pairwise similarities can be inferred from given data [20,21], so any diversity measure based on pairwise similarities is subject to an ad hoc choice of a particular similarity measure. In addition, it is unclear how heavily such an index should weigh disparity versus variety and balance [13].

More importantly, considering only pairwise similarities between types may not adequately capture total disparity, since pairwise similarities do not take into account *in which way* pairs are similar. Pairwise similarities are typically inferred by using some measure of how many features two types share from a predefined set of features. Types in a collection may then all be similar because each pair shares *the same feature*, or because each pair shares a *different* feature. Both situations could have different diversities, but have an identical similarity structure.

This paper presents a framework to measure diversity that does not rely on pairwise similarities between types. Instead, disparity is taken into account by looking at the overlap of features between types over the whole set. This is done by drawing on the concepts of α-, β- and γ-diversities from ecology [22] and the corresponding decomposition of diversity as introduced in [17], which is based on Hill numbers [16]. The result is a measure of diversity that incorporates variety, balance and disparity simultaneously, and has a natural interpretation as the 'number of compositional units' [23].

Building on this measure, I introduce the '*ABC* decomposition' that decomposes diversity into separate measures of variety, balance and disparity. This enables the study of the distinct role each of these dimensions has in different systems.[2] The proposed framework is closely related to information-theoretic measures of uncertainty, and the use of multivariate information theory shows how the measure can be extended to take into account disparity along multiple dimensions. This leads to two results regarding the diversity of types given multiple feature sets, depending on the dependence structure of the variables involved. First, diversity considering multiple feature sets becomes multiplicative when different feature sets are independent. Second, additional feature sets may be neglected in measuring diversity when one feature set is conditionally independent of the types, given another feature set.

I proceed as follows. Section 2 starts with an example of a situation where using pairwise similarities fails to quantify disparity correctly. Subsequently the concepts of β-diversity are introduced along with

---

[1]I follow the terminology used in [13], but these concepts are known by different names in different fields, for example, as 'richness', 'evenness' and 'similarity' in ecology.

[2]For example, the separate effects of variety, balance and disparity on scientific impact was studied in [8].

the main result, namely a measure of diversity that takes into account disparity as the overlap over a set of features. Section 3 then introduces a decomposition of diversity into separate measures of variety, balance and disparity. As an illustration, I apply the proposed measures to historical data in order to characterize the change in diversity of industries in the USA, taking into account disparity in terms of the occupations that industries employ. Section 4 shows how the framework can be extended to take into account multiple sets of features. I conclude with a brief discussion of the results.

# 2. Decomposing diversity

## 2.1. An example

Consider a region in which certain economic activities take place in the form of industries. These industries can be thought to consist of a certain set of inputs or features [24]. We will represent these features with letters in a set $S$, and the industries as words in set $S'$. For example, one might think of the letters as occupations required by a firm to engage in a particular industry, represented as a word. The diversity of words is determined by the number of different words (variety), their relative frequency (balance) and their similarity in terms of the letters they consist of (disparity). Adding words with similar composition of letters does not affect the diversity much, whereas adding words consisting of many new letters may greatly increase diversity.

The composition of words and letters in a region can be represented as a bipartite network as in figure 1. In the three cases shown, the variety equals 3 (there are three unique words) and the balance is maximal (the relative frequency $p_i = (1/3)$ for each word). The disparity of words is different for each of the three cases, and is determined by how the words are composed from the letters.

A common approach to quantify diversity while taking into account disparity is by considering the pairwise similarity between types [13–15,18]. Computing the pairwise similarities can be interpreted as 'projecting' the bipartite network onto a weighted network in which the nodes are the types, and the weighted edges represent the pairwise similarities in terms of the overlap in features (figure 1). Here we consider the Jaccard similarity $s_{ij}$, which gives the similarity as the number of shared features divided by the total number of features used by both types.

An example of such a measure is the Rao–Stirling diversity, which is computed as[3] [13,14]

$$\Delta = \sum_{ij} (1 - s_{ij}) p_i p_j.$$

This measure incorporates the variety by summing over all types, and the balance by taking into account the relative frequencies $p_i$. Disparity is then taken into account by weighing every pair of types by the distance between the types. This way, pairs with low similarity contribute more to the diversity than pairs with high similarity.

In the first case in figure 1, the disparity is maximal (there is no overlap of letters between words), and the Rao–Stirling diversity reduces to $\Delta = \sum_{ij} (1/3)(1/3) = (1/3)$ since $s_{ij} = 0$ for all pairs. For the other two cases, the Jaccard similarities are given by $s_{ij} = (1/5)$ for all pairs. Since the pairwise similarities are identical in both cases, any diversity measure based on these pairwise similarities will give the same diversity for both cases. Indeed, computation of Rao–Stirling diversity shows a diversity of $\Delta = \sum_{ij} (1 - (1/5))(1/3)(1/3) = (4/15)$ for both cases.

However, note that the underlying network structure in the latter two cases in figure 1 is different. While both have an identical variety and balance of words, the distribution of letters is different. In the middle case in figure 1, all words share the same letter so that every word pair is similar *in the same way*. In the latter case, every word pair shares a *different* letter, so they are similar in different ways. This leads to a different distribution of features in both cases. Out of two collections of types with the same variety and balance, the collection that represents a higher diversity of features is arguably more diverse when taking into account the disparity between types. Hence, we expect the case with a higher diversity of features in figure 1 to have a higher diversity. Since the projected networks for the middle and last case are identical, however, such a difference cannot be captured by diversity measures that are based on those pairwise similarities. This paper proposes a measure that takes into account the overlap in features over the whole collection of types, as opposed to pairwise similarities, leading to a measure that reflects the difference in composition between the two cases.

---

[3]$1 - s_{ij}$ gives the 'Jaccard distance' or dissimilarity between a pair of words.

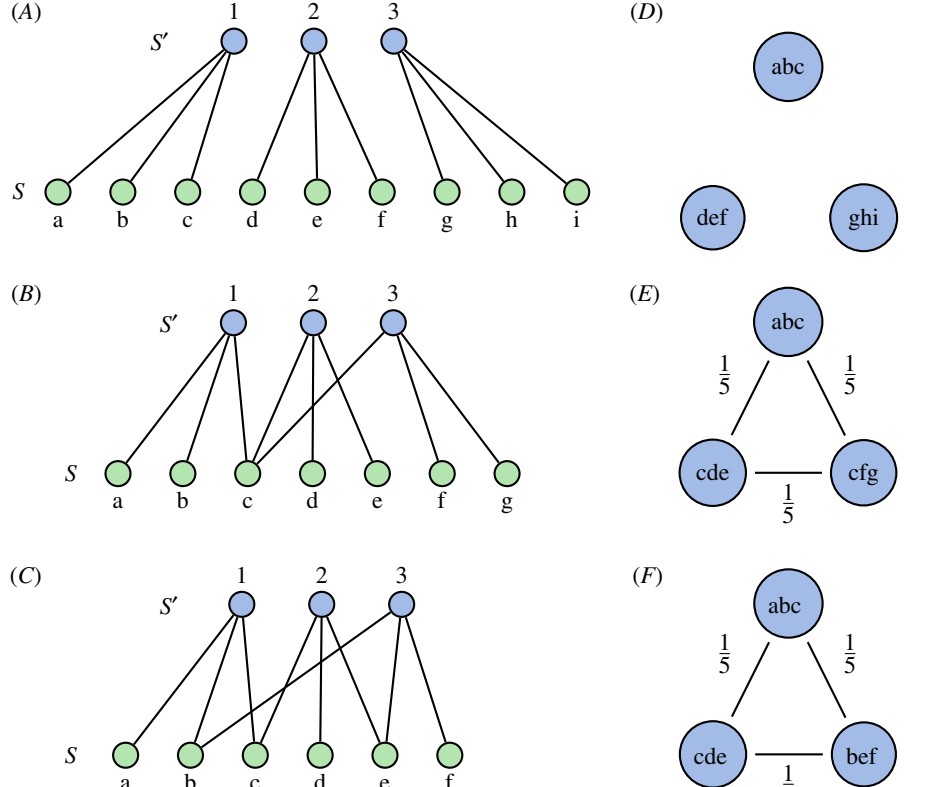

**Figure 1.** *A*, *B* and *C* show the bipartite networks as discussed in the main text. One can think of the blue nodes representing three industries (words), and the green nodes representing nine occupations (letters) that characterize the industries. *D*, *E* and *F* show the corresponding projected industry networks, in which the edge weights are given by the Jaccard similarity between industries. In *A* and *D*, there is no overlap in occupations, and pairwise similarities between industries are 0, as shown by the absence of edges in *D*. The Rao–Stirling diversity is given by $\Delta = (1/3)$. *B* and *E* show a situation where the industries use a total of seven occupations, and the similarity between each industry equals $s_{ij} = (1/5)$. *C* and *F* show a situation where only six occupations are present, and where all pairwise similarities are again $(1/5)$. Although *B* has a different distribution of occupations than *C*, their projections *E* and *F* are identical, and therefore any diversity measure based on those pairwise similarities will assign identical diversities to both cases. The Rao–Stirling diversity is given by $\Delta = (4/15)$ for both cases.

## 2.2. Hill numbers

In our measurement of diversity, we build on the framework of Hill numbers, which provides a unifying mathematical framework for the measurement of diversity when disparity is not taken into account [16,25]. Hill numbers define diversity as the inverse of a generalized weighted average of the relative frequencies of the types. In this definition, a collection is diverse if the types are on average rare, i.e. the average share of the types is low.

Hill numbers satisfy a number of axiomatic requirements for a measure of diversity, including symmetry, continuity and monotonicity in the number of species [10]. Another key property is the *replication principle*, which states that pooling together two collections that do not share any types but have equal distributions, should give a new collection with double the diversity of the original collections [16].

Hill numbers give rise to a parametric family of diversity measures, in which a parameter $q$ determines how heavily one weighs the rarity of types in a measure of diversity. For $q = 1$, rare and common species are weighed equally heavy and the Hill number equals the exponential of the Shannon entropy,

$$D(S) = e^{H(X)} = e^{-\sum_i p_i \log p_i}. \tag{2.1}$$

Here, $S$ is a collection of elements with types $i$ and relative frequencies $p_i$, and $X$ is a random variable that represents the type $i$ of a randomly drawn element from $S$. A more elaborate discussion on Hill numbers and their relation to Shannon entropy can be found in the electronic supplementary material, A. It was shown in [17] that the Hill number with $q = 1$, i.e. the exponential of the Shannon entropy, is the unique

**Table 1.** Values of the α-, β- and γ-diversities for the three examples depicted in figure 1. The average diversity of occupations within an industry, $D_\alpha(S')$ is equal in all three examples, as every industry employs three different occupations with equal weight and all industries have an equal share. The total diversity of features $D_\gamma(S)$ is given by the effective number of occupations in all industries pooled together, and differs in each case. This also leads to different values of $D_\beta(S')$. For completeness, the effective number of industries $D(S')$, representing the diversity of industries when one assumes that they are totally disparate, is also included.

| | α-diversity $D_\alpha(S')$ | β-diversity $D_\beta(S')$ | γ-diversity $D_\gamma(S)$ | eff. number $D(S')$ |
|---|---|---|---|---|
| A | 3 | 3 | 9 | 3 |
| B | 3 | 2.08 | 6.24 | 3 |
| C | 3 | 1.89 | 5.67 | 3 |

measure that satisfies all axiomatic requirements and allows for a decomposition of independent within- and between-components in the presence of groups.

Hill numbers have also been referred to as the 'true diversity' as opposed to an index, as many existing diversity indices in ecology and economics that were originally introduced based on heuristics have been shown to be a transformation of a Hill number [25]. In particular, equation (2.1) shows how the Shannon entropy, a popular *index* of diversity but which is actually a measure of uncertainty (it has units in 'bits' or 'nats'), can be transformed into a measure of diversity [25].

Furthermore, the Hill number of a collection has a clear interpretation as the 'effective number' of types, meaning that the Hill number of a collection $S$ can be interpreted as the number of types that would be present in a virtual collection $\tilde{S}$ that has maximal balance (i.e. a uniform distribution over types) and has the same diversity as $S$. In particular, for a uniform distribution, i.e. $p_i = (1/n)$ for all $i$, we have $D(S) = n$ so that the diversity equals the number of types. For any other distribution over types, the Hill number represents the equivalent number of types in a maximally balanced collection.

## 2.3. α- and β-diversities and the number of compositional units

The Hill number $D(S)$ quantifies both the variety and balance of types but not their disparity, and thus implicitly assumes that all types $i$ are maximally disparate. Here, we aim to extend this framework to include the overlap of features between types. To this end, we build on the concepts of α-, β- and γ-diversities from ecology.

Hill numbers provide a decomposition of diversity into its α- and β-components [17], which are used in ecology to describe the average within-sample diversity and the between-sample diversity, respectively [22]. For example, consider a forest in which the distribution of species is sampled in different plots. The diversity of the collection of species that consists of all plots pooled together is called the total diversity or γ-diversity. The α-diversity represents the average diversity *within* each plot. The β-diversity represents the diversity *between* each plot, reflecting the diversity that is the result of the differences in species composition between each plot.

The γ-diversity of the forest can be multiplicatively decomposed into independent α- and β-components, i.e. $D_\gamma = D_\alpha D_\beta$ [17]. In a homogeneous forest, where all plots have approximately the same species composition, the average within-plot diversity $D_\alpha$ is close to the diversity of all plots pooled together, $D_\gamma$. Hence, the between-plot diversity $D_\beta$ will be close to 1. In a heterogeneous forest on the other hand, every plot has a very different species composition and contains only a small part of the total diversity, so $D_\alpha$ is much smaller than $D_\gamma$, leading to a higher value of $D_\beta$. $D_\beta$ reflects the number of different plots needed, each with diversity $D_\alpha$, to obtain a pooled diversity of $D_\gamma$. The maximum value of $D_\beta$ is given by $D_\gamma$, corresponding to the case where every plot consists of a unique species ($D_\alpha = 1$). The β-diversity is thus bound from below by 1 and from above by $D_\gamma$.

Note that the situation described above corresponds with the example in figure 1, in which the plots represent the types of interest (words) and the species represent some characterizing features of those types (letters). In this setting, the γ-diversity gives the total diversity of features, and the α-diversity the average diversity of features within a type. The β-diversity then represents the 'between-type' diversity based on the heterogeneity of the composition of types.

The values of each diversity for the example in figure 1 are given in table 1. Since for every case each of the three words contains three letters, the α-diversity is three for all cases. The diversity of letters as

measured by the Hill number is different for each case, however, as shown by the γ-diversity. The diversity of letters is lowest for the last example. This is reflected by the β-diversity, which gives a lower number of compositional units for the case with a lower diversity of features.

The β-diversity gives the number of types with average diversity $D_\alpha$ that are needed to obtain a total diversity of features $D_\gamma$ when there would be no overlap of features between the types. It is obtained by dividing the total diversity of features, as given by the Hill number of order 1, by the average diversity of features within a type, so that

$$D_\beta(S') = \frac{D\gamma(S)}{D_\alpha(S)}. \tag{2.2}$$

It can be interpreted as a measure of the 'number of compositional units', giving the effective number of types that would be present when the types do not share any features and would be equally abundant [23]. Framing β-diversity in terms of types and features provides a measure of diversity of types that takes into account variety, balance *and* disparity as given by the overlap of features between types. As a measure of diversity, the number of compositional units satisfies all of the mathematical properties that were proposed by Leinster & Cobbold [15] that reflect a 'basic scientific intuition' about diversity. The nine properties are divided into three categories: partitioning properties, elementary properties and similarity properties (see the electronic supplementary material, B). How to compute the number of compositional units from data will be discussed using an empirical example in the following section.

## 2.4. Measuring diversity of industries

Here, the general application of the proposed diversity measure is presented using an empirical example. The aim is to quantify industrial diversity in the USA, where the distinguishing features of industries are considered to be the different occupations they employ. US census data were extracted from IPUMS-USA [26], providing a 1% sample[4] of total population in the USA for every decade from 1850 to 2010. The data contain for every person their occupation $i \in S$ and industry $j \in S'$. The used classifications consist of 269 occupation types and 147 industry types.

The data are interpreted as a weighted bipartite network as in figure 1, with nodes $i$ in the occupation layer $S$ and nodes $j$ in the industry layer $S'$. The edge weight between nodes $i$ and $j$ is given by the number of people $q_{ij}$ working in occupation $i$ and industry $j$. The strength of node $i$ is given by $q_i = \sum_j q_{ij}$ and represents the total employment in occupation $i$, and similarly $q_j$ denotes total employment in industry $j$. Normalizing the quantities $q_i$, $q_j$ and $q_{ij}$ by the total number of people $Q = \sum_{ij} q_{ij}$ gives the relative frequencies $p_{ij} = (q_{ij}/Q)$, $p_i = (q_i/Q)$ and $p_j = (q_j/Q)$, respectively. Each of the relative frequencies may in turn be interpreted as the probability distribution of a random variable that represents the occupation or industry type of a randomly sampled person, i.e. $p_i = P(X = i)$, $p_j = P(Y = j)$ and $p_{ij} = P(X = i, Y = j)$.

Using Hill numbers, the effective number of industries and occupations can be expressed as $D(S') = e^{H(Y)}$ and $D(S) = e^{H(X)}$, respectively. To obtain the effective number of occupations *within* an industry, consider the relative frequencies $p_{i|j} = (q_{ij}/q_j)$ of occupation $i$ in industry $j$. The occupational diversity of an industry $j$ is then given by

$$D(S_j) = e^{H(X|j)} = e^{-\sum_i p_{i|j} \log (p_{i|j})}.$$

The *average* within-industry diversity is then given by [17]

$$D_\alpha(S) = e^{H(X|Y)} = e^{-\sum_j p_j \sum_i p_{i|j} \log (p_{i|j})},$$

where $H(X|Y)$ is the conditional entropy of $X$ given $Y$. Finally, the *within* industry diversity $D_\beta$ follows from multiplicatively decomposing the total occupational diversity $D_\gamma(S)$ into its $\alpha$ and $\beta$ components, leading to equation (2.2) [17]. $D_\beta(S')$ can be interpreted as the effective number of industries, *discounted for the overlap in their occupational distributions*. Its units correspond to the number of industries that would be present in the case of equally distributed, non-overlapping industries, and where the $D_\alpha$ and $D_\gamma$ are the same.

Figure 2 shows the time evolution of variety, the effective number of industries $D(S')$ (taking into account variety and balance) and the number of compositional units $D_\beta(S')$ (which takes into account

---

[4]For 1980 and 1990 a 5% sample was given. Note that the analysis presented is for illustrative purposes, so further data cleaning and consistency issues are not considered here.

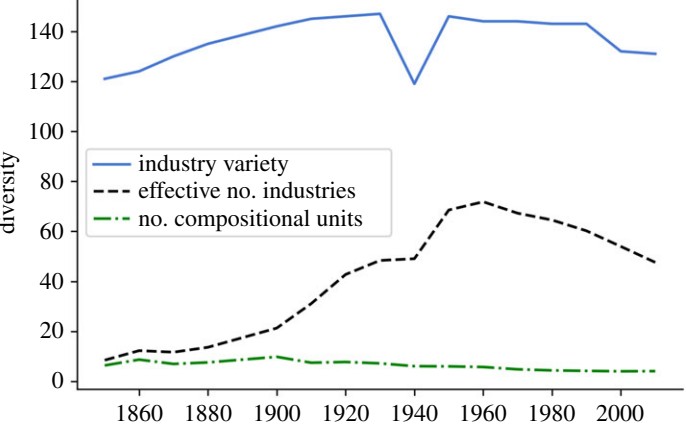

**Figure 2.** Variety, effective number and effective number of compositional units for industries. The variety of industries is approximately constant over time. The effective number of industries takes into account variety and balance and shows a hump-shaped pattern, where initially the distribution of people over industries becomes more equal reaching a diversity of 80 effective industries in 1960, where a re-concentration starts to take place. The number of compositional units takes into account the occupational overlap between industries. In 1850, the industrial diversity was equivalent to approximately 10 non-overlapping industries, which declined to approximately four compositional units in 2000.

balance, variety and disparity) of industries in the USA. The variety of industries, i.e. the number of different industry types that have at least one employee, is slightly increasing and then decreasing after 1950, with values ranging between 120 and 140 throughout the whole period. The sudden dip in variety in 1940 is unexplained, and is most likely due to data inconsistencies.

The effective number of industries $D(S')$ starts at a much lower level of an effective number of around 10 industries, showing that total employment in the 120 industry types is initially heavily concentrated in a few industries. It shows a more pronounced hump-shaped pattern, with a period of diversification in 1850–1960 in which the effective number of industries grows to around 80 industries as employment becomes more equally spread across industries, followed by a period of re-concentration after 1960. These findings are in line with work in economics that shows that countries first go through a 'diversification phase' as they develop, and then start specializing again at a later point in the development process [27].

By contrast, the number of compositional units $D_\beta(S')$ shows a pattern of steady decline since 1900, with values ranging from 10 to 4 compositional units. This means that although employment becomes more equally spread over an increasing number of industries during the diversification phase, industries become increasingly similar in terms of the occupations they employ, leading to a decreasing disparity between industries. In other words, using the notion of related variety [5], these results suggest that variety has become more related over time. This results in a decreasing number of compositional units.

It becomes clear that taking into account the different dimensions of diversity can lead to very different representations of the same data. Considering only the variety for example may lead to an overestimation of diversity, as the distribution over industries may be concentrated in only a few industries. Furthermore, when industries with similar occupational distributions are considered to be the same, the effective number of industries is an overestimation of the diversity, as some industries may be almost identical in terms of the occupations they employ.

The effect of taking into account disparity of course depends on which features are considered. That is, there might have been an increase in diversity when some other feature of industries was taken into account instead of the occupations they employ. Hence, application of these measures must be driven by the research question at hand. The interesting aspect of these measures, however, is that each dimension of diversity may show a distinct dynamic, which is not visible when considering diversity as a whole. As figure 2 shows, the number of compositional units may decrease while both the balance and variety increase. Therefore, we consider a decomposition of the number of compositional units into separate measures of variety, balance and disparity in the next section.

## 3. The 'ABC' decomposition

In order investigate the role of variety, balance and disparity in practice, separate measures are required for each. To this end, I introduce the 'ABC decomposition', which decomposes diversity into its separate

**Table 2.** Values for the effective number of industries, the number of compositional units and the variety, balance and disparity as given by the *ABC* decomposition for the three examples depicted in figure 1. Variety and balance are equal in all three examples, as every industry employs three different occupations with equal weight. The disparity differs in all three cases, and is maximal for *A*, in which there is no overlap of occupations between industries. For *B* and *C*, the measures show a lower disparity and hence a lower diversity for *C*, in which the industries are composed of less occupations.

|   | eff. number $D(S')$ | (β-)diversity $D_\beta(S')$ | variety $D_A(S')$ | balance $D_B(S')$ | disparity $D_C(S')$ |
|---|---|---|---|---|---|
| *A* | 3 | 3 | 3 | 1 | 1 |
| *B* | 3 | 2.08 | 3 | 1 | 0.69 |
| *C* | 3 | 1.89 | 3 | 1 | 0.63 |

dimensions. Since $D_\beta(S')$ is a measure of diversity incorporating all three dimensions, a multiplicative decomposition into the variety (*A*), balance (*B*) and disparity (*C*) may be obtained as

$$D_\beta(S') = D_A(S') \cdot D_B(S') \cdot D_C(S'). \tag{3.1}$$

The variety $D_A$ is given by a simple count of the number of types in $S'$, or equivalently by the Hill number of order $q = 0$ (see electronic supplementary material, A). The balance $D_B$ is computed by dividing the effective number of types in $S'$ (which takes into account both balance and variety) by the variety, leading to [16]

$$D_B(S') = \frac{D(S')}{D_A(S')} = \frac{D(S')}{n},$$

where $D_B(S')$ measures the evenness in the distribution of relative frequencies of the types. It takes values in $((1/n), 1)$, with a maximum of 1 that is attained when all relative frequencies are equal, i.e. $p_i = (1/n)$ for all types $i$ in $S$. The minimum $(1/n)$ is achieved when the proportion of all but one type is vanishingly small.

Note that the obtained components of variety and balance are not independent, since a higher variety allows for a lower balance. For example, if nearly all employment is concentrated in one out of two industries, this gives a higher balance than a situation in which nearly all employment is concentrated in one out of 100 industries. Hence $D_B(S')$ is an 'absolute' measure of balance, as opposed to a 'relative' measure that characterizes the balance *given* a certain variety [28]. An in-depth study concerning measures of balance and their (in)dependence with variety is given in [28].

Since $D_\beta(S')$, $D_A(S')$ and $D_B(S')$ are then determined, the disparity $D_C(S')$ can be obtained by dividing the number of compositional units $D_\beta(S')$ (which takes into account all three dimensions) by the effective number as

$$D_C(S') = \frac{D_\beta(S')}{D_A(S')D_B(S')} = \frac{D_\beta(S')}{D(S')} = e^{-H(Y|X)}.$$

$D_C(S')$ can be considered as the number of compositional units normalized for variety and balance, leaving a measure of disparity. It takes values in (0, 1), attaining the maximum value when none of the types have overlap in their features. The minimum is attained when all types have identical features.

It is easily verified that (3.1) holds with these definitions of $D_A(S')$, $D_B(S')$ and $D_C(S')$. The decomposition allows to study the three dimensions of diversity separately. The diversity $D_\beta(S')$ can be seen as the variety $D_A(S')$, corrected by the factors $D_B(S')$ and $D_C(S')$ which are both between 0 and 1. The variety can in turn be normalized by the total number of types in the classification considered to make it have values between 0 and 1 so that it is comparable to the balance and the disparity as a fraction of its maximum value.

Applying the *ABC* decomposition to the example in figure 1 leads to the results given in table 2. The results show, as expected, a decreasing disparity as the overlap between words increases. The decrease in disparity as the total number of letters decreases is accurately captured by the proposed measure.

Figure 3 shows the *ABC* decomposition applied to the empirical example of industries in the USA. It contains the same information as figure 2, but shows the dynamics of variety, balance and disparity separately. Since all dimensions of diversity may move independently from each other, the *ABC* decomposition can help in analysing the specific role of each dimension in different systems.

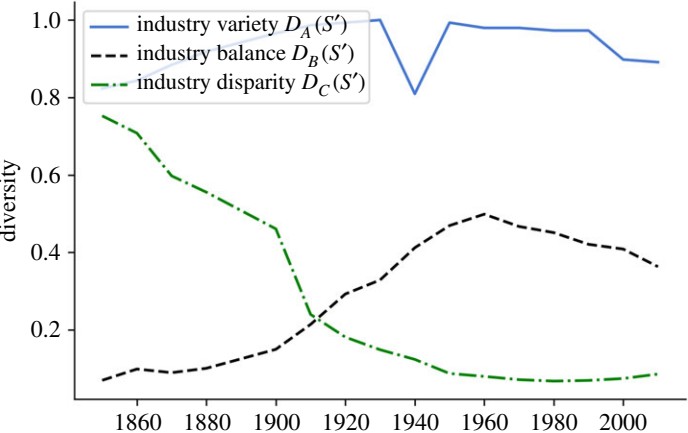

**Figure 3.** Variety, balance and distribution. Variety is normalized by the total number of possible industries in the classification, which equals 147. The variety of industries remains approximately constant over the whole period, and the balance shows diversification of industries up to 1960, followed by a period of re-concentration. The disparity shows a decline over nearly the whole period, with a slight increase since 1980.

# 4. Multivariate extensions

From the framework of Hill numbers, an interesting relation follows between diversity and the information-theoretic notion of uncertainty. In particular, the β-diversity is given by

$$D_\beta(S') = \frac{D_\gamma(S)}{D_\alpha(S)} = e^{H(X)-H(X|Y)} = e^{MI(X,Y)}, \tag{4.1}$$

where $MI(X,Y)$ denotes the mutual information between the random variables $X$ and $Y$.[5] Taking the exponential of the mutual information between random variables $X$ and $Y$ translates it into a measure of diversity of the corresponding collection $S'$, discounted for the overlap in features given by $S$. Furthermore, the additive decomposition of information-theoretic measures corresponds to a multiplicative decomposition of diversities.

## 4.1. Taking into account multiple feature sets

Here, we generalize the diversity $D_\beta$ to take into account multiple feature sets by exploiting the additive relations between multivariate information measures. For instance, returning to the example described in figure 1, we could add a feature set to each word by colour coding each letter, so that each word can be distinguished along two dimensions: its colours and its letters. Describing letters with random variable $X$, colours with random variable $Y$ and words with random variable $Z$, one can consider the joint probabilities $p_{ijk} = P(X = i, Y = j, Z = k)$ that a randomly sampled element is letter $i$, has colour $j$ and is used in word $k$. In a network representation, the joint probabilities $p_{ijk}$ can be considered as the relative frequencies of hyperlinks between nodes $i$, $j$ and $k$ in a hypergraph that connects colours, letters and words to each other.

Following equation (4.1), the diversity of words given their overlap in letters and colours is then given by

$$D_\beta^{XY}(S') = e^{H(XY)-H(XY|Z)} = e^{MI(XY,Z)},$$

where $H(XY) = -\sum_{ij} p_{ij} \log(p_{ij})$ is the Shannon entropy of the joint distribution $p_{ij}$, and the superscript in $D^{XY}(S')$ is used to indicate that diversity is taken with respect to the overlap in feature *pairs* given by $XY$. Hence, every colour–letter pair is interpreted as a distinct feature of a word.

The effect of taking into account an additional feature set on the diversity depends on the information contained by these features. In the current example, taking into account colour as a second feature set will not affect diversity much if colours and letters are highly correlated. On the other hand, diversity of words may be very high when colours and letters are independent of each other, thus capturing

[5]The mutual information is a measure of dependence between two random variables $X$ and $Y$, given by $MI(X,Y) = \sum_{ij} p_{ij} \log(p_{ij}/p_i p_j)$. It is non-negative and symmetric, and can be interpreted as the average reduction in uncertainty about the outcome of one random variable, given knowledge about the outcome of the other.

complementary information. Words that consist of the same letters may contain very different colours, and still add to the overall diversity. Mathematically, this can be seen by rewriting the β-diversity as (see electronic supplementary material, C)

$$D_\beta^{XY}(S') = e^{MI(X,Z)+MI(Y,Z)-MI(X,Y)+MI(XY|Z)},$$

from which it is clear that diversity decreases when the dependence between features, given by $MI(X, Y)$, increases. In the extreme case that letters $i$ and colours $j$ are independent, we have $MI(X, Y) = 0$ so that (see electronic supplementary material, C)

$$D_\beta^{XY}(S') = e^{MI(X,Z)+MI(Y,Z)} = D_\beta^X(S')D_\beta^Y(S'),$$

where $D_\beta^X(S')$ and $D_\beta^Y(S')$ denote the diversity of words with respect to the features described by random variables $X$ and $Y$, respectively. Thus, when feature sets are independent the diversity that takes into account feature pairs can be obtained through multiplication of the diversities that take into account each of the feature sets separately.

Results like this may be useful and relevant when estimating diversity from high-dimensional datasets containing multiple feature sets. For example, one could consider the diversity of industries by not only taking into account the occupation but also the educational profile of people employed by an industry as a distinguishing characteristic. If educational profiles and occupations are uncorrelated, this diversity equals the product of the diversities that take into account occupations and educational profiles separately.

## 4.2. Aggregation

Another interesting interpretation is to consider the types in $S'$ to be an aggregation of the features $S$. In this setting, the words are thus considered to be a specific way of aggregating letters. These words can in turn be further aggregated into sentences, effectively 'adding a layer' on top of the bipartite network depicted in figure 1. In such a setting, it follows from the current framework that the diversity of sentences depends only on the composition of words, and not on the composition of letters. The key assumption is that the two steps of aggregation are independent of each other, i.e. how words are aggregated into sentences is independent of how letters are aggregated into words.

In the situation described above, the links between letters and words are given by the joint distribution $p_{ij}$ and the links between words and sentences by $p_{jk}$, where $k$ is the index for sentences, $i$ is the index for letters and $j$ the index for words. The probability of a letter–word–sentence triplet is then given by $p_{ijk} = p_{ij}p_{k|j}$. In other words, sentences and letters are conditionally independent on knowing the word, and their joint probability is given by

$$p_{ik} = \sum_j p_{i|j}p_{k|j}p_j,$$

which implies that $MI(X, Z | Y) = 0$. The diversity of sentences given the overlap in words and letters is then equal to the diversity when considering the overlap in words only (see electronic supplementary material, C)

$$D_\beta^Z(S') = e^{MI(Z,XY)} = e^{MI(Z,Y)}. \tag{4.2}$$

Hence, when the composition of types described by $Z$ in terms of features described by $Y$ is independent of how the features $Y$ themselves are composed of other features $X$, the features $X$ are irrelevant for considering diversity of $Z$.

As an example, consider the diversity of industries in figure 2, where occupations are taken to be features of the industries. Hence, we can consider the distribution over industries as a particular way of aggregating over occupations. Similarly, occupations can be considered to be a collection of particular skills and tasks, and hence industries are, indirectly, also an aggregation of those skills and tasks. Equation (4.2) shows that as long as the composition of occupations in terms of skills and tasks is independent of the industry they are employed in, the diversity of industries is fully captured by occupations alone, and there is no need to consider skills and tasks.

# 5. Discussion

This paper presented a framework to measure diversity while taking into account variety, balance and disparity of types. The framework builds on Hill numbers [16,25] and the corresponding decomposition

of diversity into independent $\alpha$ and $\beta$ components [17]. It has a clear interpretation in terms of the 'number of compositional units' [23], and satisfies a set of basic intuitive properties for diversity measures as formulated in [15]. Contrary to current approaches [10,15,18], the measure does not rely on pairwise similarities but instead takes into account overlap of features between types over the whole set.

I have also proposed the '$ABC$' decomposition of diversity that provides a way to capture variety, balance and disparity in separate measures. Such measures may help disentangle the distinct dynamics and functional properties that different dimensions of diversity may have in different systems. In the context of economics, for example, economic development is often associated with an increase of the diversity of economic activities [24,27,29]. It is, however, an open question what the role of the individual components of diversity is in the process of economic development—as the preliminary results in the current paper show, economic development may actually go hand in hand with decreasing disparity, when disparity is measured in terms of industries and the occupations they employ.

The proposed framework reveals close connections between measuring diversity and information-theoretic measures of uncertainty. The simple additive properties of information-theoretic measures correspond to multiplicative properties of diversity measures, and enables derivation of special properties when considering multiple feature sets. These properties may provide useful tools in the analysis of high-dimensional datasets. Furthermore, the diversity measures presented here can be interpreted as centrality measures on bipartite networks, or hypergraphs in the multivariate case. In this sense, the β-diversity captures structural properties of the network. Application of these measures may also be extended to directed networks (e.g. input–output tables in economics [30]), as any directed network may be interpreted as a bipartite network.

The current paper also leaves some open challenges that have not been addressed. First, there is the issue of the estimation of the proposed diversity measures from data, and finding a measure of precision of this estimate. A promising way forward is to use a Bayesian framework as in [31,32]. These works provide closed-form solutions for the moments of the posterior distribution of information-theoretic quantities like the Shannon entropy and mutual information, given the data and a prior distribution for the probabilities $p_{ij}$. Such an approach should extend in a straightforward way to the exponentials of those quantities (which are our measures of diversity). In this way, an estimate can be obtained along with a 'Bayesian error bar' that shows the precision of that estimate given the data and a prior distribution, for example, showing a lower precision estimate when the number of observations is low [31]. A major challenge in applying such an approach for the estimation of diversity is to find a suitable prior for the joint distribution of types and features in the situation at hand. Implementing this Bayesian approach in order to provide unbiased estimates of diversity with their corresponding error bars is a topic for future research.

A second line for future investigation is to further examine the relationship between the two alternative approaches to include disparity into diversity using Hill numbers: including pairwise similarities directly into Hill numbers as in [15,18], or—as in the current paper—using α- and β-diversities instead. On the one hand, pairwise similarities between types may not adequately capture total disparity since they do not take into account in which way pairs are similar. On the other hand, taking into account the full distribution of features requires more data in order to estimate the joint distribution of types and features. Furthermore, it may be challenging to find variables that are readily interpretable as features of the types of interest. In situations where data are limited, an approach based on pairwise similarities may be preferable.

It is worth noting, however, that both approaches are not mutually exclusive. In particular, Chiu *et al.* [18] show that their measure of diversity, which generalizes Hill numbers to include pairwise similarities, allows for a decomposition into $\alpha$ and $\beta$ components. Their $\beta$ component then gives the number of compositional units while taking into account a given set of pairwise similarities between the features. This highlights the fact that both approaches provide alternative operationalizations of the concept of disparity that can even be used in tandem. In practice, selection of a method to measure diversity requires theory-driven justification, and should be guided by data availability and the research question at hand.

Data accessibility. The data and code that were used for this study are available at https://github.com/aljevandam/diversity_decomposition.

Competing interests. I declare no competing interests.

Funding. This work was funded by the Netherlands Organisation for Scientific Research (NWO) under the Vici scheme, number 453-14-014, and benefited from support from the Swaantje Mondt travel fund.

Acknowledgements. I thank Andres Gomez-Lievano, Koen Frenken and Frank Neffke for their valuable feedback and comments.

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
