## [Reviewer comments · Royal Society Open Science]

Review History

RSOS-190452.R0 (Original submission)

Review form: Reviewer 1

Is the manuscript scientifically sound in its present form?

Yes

Are the interpretations and conclusions justified by the results?

Yes

Is the language acceptable?

No

Is it clear how to access all supporting data?

No

Do you have any ethical concerns with this paper?

No

Have you any concerns about statistical analyses in this paper?

No

Recommendation?

Major revision is needed (please make suggestions in comments)

Comments to the Author(s)

I found the generalization of the Hill numbers interesting, and the approach coherent. A few comments.

1. For your real example, I found it difficult to see how your analyses might contribute to a better understanding than already exists concerning industries. In particular, I found it difficult to relate the estimates to what they imply about industry over that period. This may reflect my lack of experience of measuring diversity in this context.
2. Related to the above point, I feel that inference is limited given that there is no attempt to quantify the precision of your diversity measure or its components. It seems to be a common feature of many diversity papers that measures are estimated, yet no attempt to quantify the precision of those estimates is made! You describe apparent changes over time but with no basis for assessing whether those changes are real or simply random (possibly correlated) fluctuations.
3. Concerning fig 1, you state 'the collection that consists of more different features is arguably more diverse'. However, you could argue that counts of 1,2,2,1,2,1 are more even than 1,1,3,1,1,1,1, and different ways of measuring diversity could legitimately lead to different conclusions. While your approach is mathematically appealing and offers nice interpretation, it is not the unique, and probably not the optimal, way to quantify diversity - in fact, I don't believe there can be a single optimal approach. Different objectives will suggest different ways to measure diversity.

A final comment is that the draft could do with a final check - there are many typos!

Review form: Reviewer 2**Is the manuscript scientifically sound in its present form?**

Yes

Are the interpretations and conclusions justified by the results?

No

Is the language acceptable?

Yes

Is it clear how to access all supporting data?

Yes

Do you have any ethical concerns with this paper?

No

Have you any concerns about statistical analyses in this paper?

I do not feel qualified to assess the statistics

Recommendation?

Accept with minor revision (please list in comments)

Comments to the Author(s)

The author studies the measurement of diversity based on three components: variety, balance and disparity. He proposes a new approach based on similarities of features.

Although I did not check all mathematical formulae, I am convinced that the approach suggested by the author is worth investigating.

Some remarks

p.2 line 31-33. I was convinced that similarity as used in diversity studies always referred to the same feature, but the author claims that in the past, when calculating pairwise similarity, this term has been used for different features, i.e. in pairwise similarity comparisons, different features have been used. Is this true?

I expected that section 2(c) would end with a formula, or stepwise method, for calculating diversity. Now, this is hidden in the example 2(d).

In section 3, about the ABC decomposition, nothing is said about the alpha-diversity factor. About the D(B) factor in this decomposition. The author does not show that this factor is really variety-independent, i.e. does the same balance and different variety values, always lead to the same D(B) factor? This observation brings me to another remark. Leinster and Cobbold have a list of properties which their measure meet. They provide mathematical proofs of this. It is not clear to me, which of these properties still hold for the measure provided by the author.

A general observation. Other colleagues have proposed a decomposition of diversity, such as (Chiu & Chao, 2014) and Leydesdorff et al. (2019a), see also Leydesdorff et al. (2019b). The different influences of variety, balance and disparity on diversity have been studied in (Wang et al., 2015).

I think these studies should be mentioned and their relation with the author's work discussed.

Chiu C-H, Chao A (2014) Distance-Based Functional Diversity Measures and Their

Decomposition: A Framework Based on Hill Numbers. PLoS ONE 9(7): e100014.

doi:10.1371/journal.pone.0100014

Leydesdorff, L., Wagner, C. S., & Bornmann, L. (2019a). Interdisciplinarity as diversity in citation patterns among journals: Rao-Stirling diversity, relative variety, and the Gini coefficient. *Journal of Informetrics*, 13(1), 255–264.

Leydesdorff, L., Wagner, C. S., & Bornmann, L. (2019b). Correspondence. Diversity measurement: Steps towards the measurement of interdisciplinarity? *Journal of Informetrics* (in press; available on the journal's website).

Wang J, Thijs B, Glänzel W (2015) Interdisciplinarity and Impact: Distinct Effects of Variety, Balance, and Disparity. PLoS ONE 10(5): e0127298.

<https://doi.org/10.1371/journal.pone.0127298>

Typos:

p.3 line 7: seperate MUST BE separate

p.3 line 16 takes place MUST BE take place

p.3 line 32 wighted MUST BE weighted

p.3 line 47 simlairities MUST BE similarities

p.3 line 52 in middle MUST BE for the middle

p. 12 line 15 strutural MUST BE structural

Reference 27 is incomplete

Decision letter (RSOS-190452.R0)

07-May-2019

Dear Mr van Dam,

The editors assigned to your paper ("Diversity and its decomposition into variety, balance and disparity") have now received comments from reviewers. We would like you to revise your paper in accordance with the referee and Associate Editor suggestions which can be found below (not including confidential reports to the Editor). Please note this decision does not guarantee eventual acceptance.

Please submit a copy of your revised paper before 30-May-2019. Please note that the revision deadline will expire at 00.00am on this date. If we do not hear from you within this time then it will be assumed that the paper has been withdrawn. In exceptional circumstances, extensions may be possible if agreed with the Editorial Office in advance. We do not allow multiple rounds of revision so we urge you to make every effort to fully address all of the comments at this stage. If deemed necessary by the Editors, your manuscript will be sent back to one or more of the original reviewers for assessment. If the original reviewers are not available, we may invite new reviewers.

- Data accessibility

If you wish to submit your supporting data or code to Dryad (<http://datadryad.org/>), or modify your current submission to dryad, please use the following link:
<http://datadryad.org/submit?journalID=RSOS&manu=RSOS-190452>

- **Competing interests**

- **Authors' contributions**

- **Acknowledgements**

- **Funding statement**

Kind regards,

Andrew Dunn

on behalf of Professor Len Thomas (Associate Editor) and Mark Chaplain (Subject Editor)
openscience@royalsociety.org

Associate Editor's comments (Professor Len Thomas):

Associate Editor: 1

Comments to the Author:

Both reviewers were positive about the ideas in your manuscript, so I am happy to recommend acceptance after revision. Both made a number of suggestions; please detail how you have dealt with these in your re-submission cover letter. Please pay special attention to the following points:

- Please take a more balanced approach to extolling the virtues of this new method. As one reviewer points out, there is probably no one best measure of diversity - each has positive and negative sides. Please mention these in your revisions.

- Please do try to deal with the reviewer's point about precision measurement - how might that be achieved? You may decide not to implement your suggestion, but it should be clear to the reader how they could do it; please do implement your suggestion if you can, however.

- Please check carefully for typos and other editorial issues before resubmitting. One reviewer made some specific suggestions, but there are others they did not mention.

more tempered approach to extolling the virtues - advantages and disadvantages

precision

typos

Comments to Author:

Reviewers' Comments to Author:

Reviewer: 1

Comments to the Author(s)

I found the generalization of the Hill numbers interesting, and the approach coherent. A few comments.

1. For your real example, I found it difficult to see how your analyses might contribute to a better understanding than already exists concerning industries. In particular, I found it difficult to relate the estimates to what they imply about industry over that period. This may reflect my lack of experience of measuring diversity in this context.

2. Related to the above point, I feel that inference is limited given that there is no attempt to quantify the precision of your diversity measure or its components. It seems to be a common feature of many diversity papers that measures are estimated, yet no attempt to quantify the precision of those estimates is made! You describe apparent changes over time but with no basis for assessing whether those changes are real or simply random (possibly correlated) fluctuations.

3. Concerning fig 1, you state 'the collection that consists of more different features is arguably more diverse'. However, you could argue that counts of 1,2,2,1,2,1 are more even than 1,1,3,1,1,1,1, and different ways of measuring diversity could legitimately lead to different conclusions. While your approach is mathematically appealing and offers nice interpretation, it is not the unique, and probably not the optimal, way to quantify diversity - in fact, I don't believe there can be a single optimal approach. Different objectives will suggest different ways to measure diversity.

A final comment is that the draft could do with a final check - there are many typos!

Reviewer: 2

Comments to the Author(s)

The author studies the measurement of diversity based on three components: variety, balance and disparity. He proposes a new approach based on similarities of features.

Although I did not check all mathematical formulae, I am convinced that the approach suggested by the author is worth investigating.

Some remarks

p.2 line 31-33. I was convinced that similarity as used in diversity studies always referred to the same feature, but the author claims that in the past, when calculating pairwise similarity, this term has been used for different features, i.e. in pairwise similarity comparisons, different features have been used. Is this true?

I expected that section 2(c) would end with a formula, or stepwise method, for calculating diversity. Now, this is hidden in the example 2(d).

In section 3, about the ABC decomposition, nothing is said about the alpha-diversity factor. About the D(B) factor in this decomposition. The author does not show that this factor is really variety-independent, i.e. does the same balance and different variety values, always lead to the same D(B) factor? This observation brings me to another remark. Leinster and Cobbold have a list of properties which their measure meet. They provide mathematical proofs of this. It is not clear to me, which of these properties still hold for the measure provided by the author.

A general observation. Other colleagues have proposed a decomposition of diversity, such as (Chiu & Chao, 2014) and Leydesdorff et al. (2019a), see also Leydesdorff et al. (2019b). The different influences of variety, balance and disparity on diversity have been studied in (Wang et al., 2015).

I think these studies should be mentioned and their relation with the author's work discussed.

Chiu C-H, Chao A (2014) Distance-Based Functional Diversity Measures and Their

Decomposition: A Framework Based on Hill Numbers. PLoS ONE 9(7): e100014.

doi:10.1371/journal.pone.0100014

Leydesdorff, L., Wagner, C. S., & Bornmann, L. (2019a). Interdisciplinarity as diversity in citation patterns among journals: Rao-Stirling diversity, relative variety, and the Gini coefficient. *Journal of Informetrics*, 13(1), 255–264.

Leydesdorff, L., Wagner, C. S., & Bornmann, L. (2019b). Correspondence. Diversity measurement: Steps towards the measurement of interdisciplinarity? *Journal of Informetrics* (in press; available on the journal's website).

Wang J, Thijs B, Glänzel W (2015) Interdisciplinarity and Impact: Distinct Effects of Variety, Balance, and Disparity. PLoS ONE 10(5): e0127298.

<https://doi.org/10.1371/journal.pone.0127298>

Typos:

p.3 line 7: seperate MUST BE separate

p.3 line 16 takes place MUST BE take place

p.3 line 32 wighted MUST BE weighted

p.3 line 47 simlairities MUST BE similarities

p.3 line 52 in middle MUST BE for the middle

p. 12 line 15 strutural MUST BE structural

Reference 27 is incomplete

Author's Response to Decision Letter for (RSOS-190452.R0)

See Appendix A.

RSOS-190452.R1 (Revision)

Review form: Reviewer 1

Is the manuscript scientifically sound in its present form?

Yes

Are the interpretations and conclusions justified by the results?

Yes

Is the language acceptable?

Yes

Do you have any ethical concerns with this paper?

No

Recommendation?

Accept as is

Comments to the Author(s)

I find the methods developed in this paper interesting and potentially very useful. I find the lack of precision measures disappointing - too many authors of diversity papers draw conclusions about temporal changes without ever checking whether the estimated changes might be just random fluctuation. I accept that the changes over the timescale you consider are much greater than could be explained by chance, but I would have liked to see best practice here! I appreciate the additional discussion of what your results mean in the case of your example, and accept that a more inciteful interpretation is inappropriate given the purpose of this paper. I do wonder whether the choice of example is optimal for bringing your methods to the attention of the potential user community, but perhaps if there is a follow-up paper, in which for example methods to quantify precision are developed, then the methods will be of greater value to the user community, and a biodiversity example might be used to gain a wider audience.

Review form: Reviewer 2

Is the manuscript scientifically sound in its present form?

Yes

Are the interpretations and conclusions justified by the results?

Yes

Is the language acceptable?

Yes

Do you have any ethical concerns with this paper?

No

Recommendation?

Accept as is

Comments to the Author(s)

I am satisfied with the revision

Decision letter (RSOS-190452.R1)

24-Jun-2019

Dear Mr van Dam,

I am pleased to inform you that your manuscript entitled "Diversity and its decomposition into variety, balance and disparity" is now accepted for publication in Royal Society Open Science.

Kind regards,

Alice Power

Editorial Coordinator

on behalf of Professor Len Thomas (Associate Editor) and Mark Chaplain (Subject Editor)

Associate Editor Comments to Author (Professor Len Thomas):

Thanks for making these revisions - I have recommended the manuscript is now acceptable as is. For your information, below are some sensible comments from reviewer 2 that you may wish to bear in mind for future work.

Reviewer comments to Author:

Reviewer: 2

Comments to the Author(s)

I am satisfied with the revision

Reviewer: 1

Comments to the Author(s)

I find the methods developed in this paper interesting and potentially very useful. I find the lack of precision measures disappointing - too many authors of diversity papers draw conclusions about temporal changes without ever checking whether the estimated changes might be just random fluctuation. I accept that the changes over the timescale you consider are much greater than could be explained by chance, but I would have liked to see best practice here! I appreciate the additional discussion of what your results mean in the case of your example, and accept that a more inciteful interpretation is inappropriate given the purpose of this paper. I do wonder whether the choice of example is optimal for bringing your methods to the attention of the potential user community, but perhaps if there is a follow-up paper, in which for example methods to quantify precision are developed, then the methods will be of greater value to the user community, and a biodiversity example might be used to gain a wider audience.

Appendix A

Dear editor and reviewers,

Thank you for your kind comments and suggestions. I have tried to deal with them as best as I could. Below, you will find how I revised the paper according to every comment.

Associate Editor's comments (Professor Len Thomas):

Associate Editor: 1

Comments to the Author:

Both reviewers were positive about the ideas in your manuscript, so I am happy to recommend acceptance after revision. Both made a number of suggestions; please detail how you have dealt with these in your re-submission cover letter. Please pay special attention to the following points:

- Please take a more balanced approach to extolling the virtues of this new method. As one reviewer points out, there is probably no one best measure of diversity - each has positive and negative sides. Please mention these in your revisions.

Abstract – toned down and reformulated some sentences.

Overall, I have tried to tone down statements regarding the virtues of the method. Also, I included a paragraph in the Discussion on persisting challenges and the relation to existing approaches.

- Please do try to deal with the reviewer's point about precision measurement - how might that be achieved? You may decide not to implement your suggestion, but it should be clear to the reader how they could do it; please do implement your suggestion if you can, however.

This could be achieved using a Bayesian approach as formulated by e.g. Wolpert & Wolf (1995) and Hutter & Zaffalon (2005). I added to a paragraph on this to the Discussion. Implementation of this is in my opinion beyond the scope of this paper, as the points I wish to make are mainly theoretical. The section in the Discussion is meant to provide a starting point for future empirical work.

- Please check carefully for typos and other editorial issues before resubmitting. One reviewer made some specific suggestions, but there are others they did not mention.

I have made my best effort to check for and correct any typos.

more tempered approach to extolling the virtues - advantages and disadvantages

precision

typos

Comments to Author:

Reviewers' Comments to Author:

Reviewer: 1

Comments to the Author(s)

I found the generalization of the Hill numbers interesting, and the approach coherent. A few comments.

1. For your real example, I found it difficult to see how your analyses might contribute to a better understanding than already exists concerning industries. In particular, I found it difficult to relate the estimates to what they imply about industry over that period. This may reflect my lack of experience of measuring diversity in this context.

I rewrote part of the discussion of the figure and added the Imbs & Wacziarg (2003) reference to provide context. It should become clear that taking into account disparity may give additional insight into industrial dynamics, and put into a new light previous results regarding the relation between diversity and economic development. I also include a reference to the concept of related variety (Frenken et al., 2007).

2. Related to the above point, I feel that inference is limited given that there is no attempt to quantify the precision of your diversity measure or its components. It seems to be a common feature of many diversity papers that measures are estimated, yet no attempt to quantify the precision of those estimates is made! You describe apparent changes over time but with no basis for assessing whether those changes are real or simply random (possibly correlated) fluctuations.

An estimate for the precision of inferred diversities could be achieved using a Bayesian approach as formulated by e.g. Wolpert & Wolf (1995) and Hutter & Zaffalon (2005). I added a paragraph on this to the Discussion. Implementation of this is in my opinion beyond the scope of this paper, as the points I wish to make are mainly theoretical. The section in the Discussion is meant to provide a starting point for future empirical work.

3. Concerning fig 1, you state 'the collection that consists of more different features is arguably more diverse'. However, you could argue that counts of 1,2,2,1,2,1 are more even than 1,1,3,1,1,1, and different ways of measuring diversity could legitimately lead to different conclusions. While your approach is mathematically appealing and offers nice interpretation, it is not the unique, and probably not the optimal, way to quantify diversity - in fact, I don't believe there can be a single optimal approach. Different objectives will suggest different ways to measure diversity.

The reviewer is right in their remark – though the variety of features is lower, the balance of features is higher for the last case in the example. I know do not state which case has a higher diversity when introducing the example – only that one would expect a higher diversity for a higher diversity of features. I then come back to this point at the end of Section 2c), where I state the the diversity *as measured by the Hill number* is lowest in the latter case of the example.

I also changed this in the caption of Figure 1.

A final comment is that the draft could do with a final check - there are many typos!

I have made my best effort to check for and correct any typos.

Reviewer: 2

Comments to the Author(s)

The author studies the measurement of diversity based on three components: variety, balance and disparity. He proposes a new approach based on similarities of features.

Although I did not check all mathematical formulae, I am convinced that the approach suggested by the author is worth investigating.

Some remarks

p.2 line 31-33. I was convinced that similarity as used in diversity studies always referred to the same feature, but the author claims that in the past, when calculating pairwise similarity, this term has been used for different features, i.e. in pairwise similarity comparisons, different features have been used. Is this true?

Here I mean similarity in terms of overlap of features, where the features are represented as nodes of the 'feature layer' in a bipartite network. Thus in the case of words and letters, each letter is a feature.

The confusion here may be due to the mixed use of the term 'features' in the paper. I now use it consistently, where features are the nodes within one feature set. The section on multivariate extensions thus considers multiple features sets. Using this terminology, the reviewer is right in that previous work typically considers one feature set to define pairwise similarities between types.

I reformulated the lines that were on p31-33.

I expected that section 2(c) would end with a formula, or stepwise method, for calculating diversity. Now, this is hidden in the example 2(d).

I have tried to address this point without changing the structure of the paper, since I want to keep the paper accessible on an 'intuitive level' up to section 2d), without introducing notation. I now include equation 2.2 into section 2c) along with paragraph that summarizes the intuition of the proposed approach.

In section 3, about the ABC decomposition, nothing is said about the alpha-diversity factor.

Indeed, section three is about a decomposition of the number of compositional units into variety, balance and disparity. The alpha-diversity factor is only used before, for computation of the beta-diversity (number of compositional units).

About the D(B) factor in this decomposition. The author does not show that this factor is really variety-independent, i.e. does the same balance and different variety values, always lead to the same D(B) factor?

D(A) and D(B) are not independent, and they are not claimed to be. I added a paragraph in which this is clarified, along with a short motivation on why this is ok. How to measure balance is a non-trivial and an interesting question in itself (what would 'the same balance' mean in the reviewer's comment?). I added reference Jost (2010), which provides an in-depth analysis of measures of balance and their interpretation (including 'absolute' versus 'relative' measures of balance).

This observation brings me to another remark. Leinster and Cobbold have a list of properties which their measure meet. They provide mathematical proofs of this. It is not clear to me, which of these properties still hold for the measure provided by the author.

I have included in the appendix Section B that shows that the number of compositional units as a measure of diversity satisfies all properties that are posed in Leinster and Cobbold (2012).

I also make mention of this in the main text at the end of section 2c) and in the Discussion.

A general observation. Other colleagues have proposed a decomposition of diversity, such as (Chiu & Chao, 2014) and Leydesdorff et al. (2019a), see also Leydesdorff et al. (2019b). The different influences of variety, balance and disparity on diversity have been studied in (Wang et al., 2015).

I think these studies should be mentioned and their relation with the author's work discussed. Chiu C-H, Chao A (2014) Distance-Based Functional Diversity Measures and Their Decomposition: A Framework Based on Hill Numbers. PLoS ONE 9(7): e100014. doi:10.1371/journal.pone.0100014

This work is now mentioned in the introduction along with Leinster & Cobbold (2012) as they both are efforts to place Rao-Stirling/pairwise similarity-based diversity within the framework of Hill numbers.

I also mention this study in the Discussion, where I emphasize that the current approach (using alpha and beta diversity to infer disparity) is complementary to the approach of Chiu & Chao (2014) (including pairwise similarities into Hill number), and that both could in principle be applied at the same time.

Leydesdorff, L., Wagner, C. S., & Bornmann, L. (2019a). Interdisciplinarity as diversity in citation patterns among journals: Rao-Stirling diversity, relative variety, and the Gini coefficient. *Journal of Informetrics*, 13(1), 255–264.

Leydesdorff, L., Wagner, C. S., & Bornmann, L. (2019b). Correspondence. Diversity measurement: Steps towards the measurement of interdisciplinarity? *Journal of Informetrics* (in press; available on the journal's website).

Leydesdorff (2019a) is now mentioned in the introduction as alternative approach to the diversity problem, namely composing an index of diversity from separate measures of variety, balance and disparity.

Wang J, Thijs B, Glänzel W (2015) Interdisciplinarity and Impact: Distinct Effects of Variety, Balance, and Disparity. PLoS ONE 10(5): e0127298. <https://doi.org/10.1371/journal.pone.0127298>

This work is now mentioned in the introduction as an empirically-driven / statistical approach to obtain separate components of diversity from a collection of indices, using factor analysis.

Typos:

p.3 line 7: seperate MUST BE separate

p.3 line 16 takes place MUST BE take place
p.3 line 32 wighted MUST BE weighted
p.3 line 47 similairities MUST BE similarities
p.3 line 52 in middle MUST BE for the middle
p. 12 line 15 strutural MUST BE structural
Reference 27 is incomplete

Typos are fixed – ref27 is changed to be a published version of the working paper – Hutter & Zaffalon (2005).